# Comparisons between dipeptidyl peptidase-4 inhibitors and other classes of hypoglycemic drugs using two distinct biomarkers of pancreatic beta-cell function: A meta-analysis

**Masahiro Takahashi** [ID]*, **Misa Shibasaki, Hirotoshi Echizen, Akifumi Kushiyama** [ID]

Department of Pharmacotherapy, Meiji Pharmaceutical University, Kiyose City, Tokyo, Japan

* t-masa@my-pharm.ac.jp

## Abstract

**Data Availability Statement:** All relevant data are within the paper and its Supporting Information files.

### Background and objective

Dipeptidyl peptidase-4 (DPP-4) inhibitors have been suggested to have pancreatic beta-cell preserving effect according to studies using homeostatic model of assessment for beta-cell function (HOMA-β). However, whether HOMA-β is a suitable biomarker for comparisons between hypoglycemic drugs with different mechanisms of action remains unclear. Therefore, we conducted a meta-analysis to compare the effects of DPP-4 inhibitors and other classes of hypoglycemic drugs on HOMA-β and proinsulin-to-insulin ratio (PIR).

### Methods

We searched MEDLINE, CENTRAL, and Ichushi-web for the period of 1966 to May 2020. We collected randomized, controlled clinical trials in patients with type 2 diabetes mellitus comparing DPP-4 inhibitors and other classes of hypoglycemic agents [α-glucosidase inhibitors (α-GIs), glucagon-like peptide-1 (GLP-1) analogues, metformin, sodium-glucose cotransporter 2 (SGLT2) inhibitors, sulfonylureas, or thiazolidinediones]. Weighted mean differences and 95% confidence intervals of changes in HOMA-β or PIR during study periods were calculated for pairwise comparisons.

### Results

Thirty-seven and 21 relevant trials were retrieved for comparisons of HOMA-β and PIR, respectively. HOMA-β and PIR consistently showed superiority of DPP-4 inhibitors compared with α-GIs. Both biomarkers consistently supported inferiority of DPP-4 inhibitors compared with GLP-1 analogues. However, PIR showed inferiority of DPP-4 inhibitors compared with metformin, and superiority compared with SGLT2 inhibitors, whereas HOMA-β showed no significant differences between DPP-4 inhibitors and the two other agents.

**Funding:** The authors received no specific funding for this work.

**Competing interests:** The authors have declared that no competing interests exist.

## Conclusion

DPP-4 inhibitors appear to be superior to α-GIs but inferior to GLP-1 analogues in preservation of beta-cell function assessed by either HOMA-β or PIR. DPP-4 inhibitors seem to be superior to SGLT2 inhibitors but inferior to metformin on islet function assessed only by PIR. Because HOMA-β and PIR may indicate different aspects of beta-cell function, results of beta-cell function preserving effects of hypoglycemic agents should be interpreted with caution.

## Introduction

Patients with type 2 diabetes mellitus (DM) may exhibit impaired insulin secretion and insulin resistance. While multifaceted pathophysiological abnormalities of pancreatic islet cells are known to be involved in progressive deterioration of beta-cell mass and function in DM patients [1–4], debates continue on whether some hypoglycemic drugs may have pancreatic beta-cell function preserving effects. There is a possibility that dipeptidyl peptidase-4 (DPP-4) inhibitors may slow deterioration of beta-cell function in patients with type 2 DM, since administration of these drugs increased or restored the pancreatic beta-cell mass in mice and rats [5–7]. At present, however, it remains unclear whether DPP-4 inhibitors have advantage over other hypoglycemic drugs by exhibiting pancreatic beta-cell function preserving effect.

While many biomarkers have been proposed for quantifying *in vivo* pancreatic beta-cell function, the homeostatic model of assessment of beta-cell function (HOMA-β) has been employed most frequently in epidemiological studies [8]. Recently, the proinsulin-to-insulin ratio (PIR) has also been used in many studies as a biomarker of beta-cells [9, 10]. While HOMA-β may be associated with insulin secretion or beta-cell mass [11], PIR may reflect the efficiency of proinsulin processing within beta-cells [12]. As a result, it is of interest to study whether the two biomarkers show similar or different changes over time after initiation of hypoglycemic drugs with different mechanisms of action in patients with type 2 DM.

Previous studies using mainly HOMA-β as biomarker demonstrated that DPP-4 inhibitors may have greater pancreatic beta-cell function preserving effect compared with placebo in patients with type 2 DM [10, 13–16]. Many pair-wise comparisons of DPP-4 inhibitors with α-glucosidase inhibitors (α-GIs), metformin or sulfonylureas using various biomarkers have been reported [17–20]. However, none of the previous studies in patients with type 2 DM compared the pancretic beta-cell function preserving effect between DPP-4 inhibitors and other classes of hypoglycemic drugs using both HOMA-β and PIR as biomarkers. Here, we present our results of a meta-analysis that cast doubt about interpretationg epidemiologic data based on assessment by HOMA-β alone.

## Materials and methods

The present study was performed according to the Preferred Reporting Items of Systematic Reviews and Meta-Analyses (S1 Table) [21].

### Curation of literature

We searched for relevant studies using MEDLINE (from 1966 to May 2020), Cochrane Central Register of Controlled Trials (CENTRAL, Issue 5 of 12, May 2020), and Ichushi-web (Japan Medical Abstracts Society, from 1983 to May 2020). The search formula consisted of generic

names of the seven DPP-4 inhibitors (alogliptin, anagliptin, linagliptin, saxagliptin, sitagliptin, teneligliptin, and vildagliptin) AND (diabetes mellitus [all fields]) AND (Randomized Controlled Trial [publication type]). The detailed search strategies for each database were shown in S2 Table. Further search was performed using information available from the references of the retrieved studies, where necessary.

Inclusion criteria were: (1) studies were written in English or Japanese; (2) patients with type 2 DM were studied; (3) changes in HOMA-β and/or PIR from the beginning to the end of study were available; (4) comparisons of biomarkers were made between one of the seven DPP-4 inhibitors and another class of hypoglycemic agent [α-GIs, GLP-1 analogues, metformin, sodium-glucose cotransporter 2 (SGLT2) inhibitors, sulfonylureas, or thiazolidinediones]; and (5) studies had a prospective and randomized design. When biomarkers were measured more than once during the study period, or interim and final reports of a study were published independently, we only used the data having the longest follow-up period for that study. Inclusion of individual studies in the meta-analysis was decided by two independent reviewers (MT and MS) according to the above inclusion criteria. In the case of disagreement between two reviewers on the decision of inclusion, the difference was resolved by discussion.

## Biomarkers

HOMA-β (%) was calculated according to the following equation: (20 × insulin [μU/mL])/ (glucose [mmol/L]– 3.5) [22], or by computer program [23]. HOMA-β values are known to be reduced in patients with type 2 DM, and the reduction is considered to be associated with pancreatic beta-cell dysfunction. PIR was calculated as serum proinsulin level divided by insulin level. When PIR was given as (pmol/L)/(μU/mL), the value was divided by 7.175 [24] and converted to dimensionless value. Elevated PIR has also been known to be another biomarker of beta-cell dysfunction.

## Data analysis

Mean and standard deviation (SD) of within-patient changes in HOMA-β and/or PIR from baseline to the end of study were computed from the curated data (numerical data in tables or text). When only graphical presentations of the data were available, relevant data points were converted to digital data using UnGraph ver. 5 digitizer program (BIOSOFT, Cambridge, United Kingdom), which gives X and Y coordinates of lines and points on scanned images. When multiple doses of a DPP-4 inhibitor were given to different groups of patients, the data obtained from patients given doses that are approved in Japan were used for analysis. When data of biomarkers at baseline and at the end of study were available only as group mean values (not as within-patient changes), we substituted within-group changes by within-patient changes. We also estimated SD of the within-patient changes according to the equation reported elsewhere [25].

$$SD = \sqrt{\frac{SD_{before}^2}{N_{before}} + \frac{SD_{after}^2}{N_{after}}} \times \sqrt{N_{before} + N_{after}}$$

where $N_{before}$ and $N_{after}$ are the numbers of patients before and after intervention, and $SD_{before}$ and $SD_{after}$ are the SD of the group before and after intervention, respectively. When no SD was given in the original study and only 95% CI was given, we estimated the corresponding SD values according to the equation given below [26]:

$$SD = \sqrt{N} \times (\text{upper limit} - \text{lower limit}) \times \frac{1}{3.92}$$

For statistical analysis of meta-analysis, we calculated weighted mean differences (WMDs) of HOMA-β or PIR for DPP-4 inhibitor groups versus comparator drug groups, and the 95% confidence intervals (CIs).

We compared the pancreatic beta-cell function preserving effects of various hypoglycemic drugs using DPP-4 inhibitors as reference. Changes in mean HOMA-β during the study period obtained from hypoglycemic drugs other than DPP-4 inhibitors (such as GLP-1 analogues) were subtracted from the value obtained from DPP-4 inhibitors. Consequently, when the calculated value was positive, the DPP-4 inhibitor was considered to have greater beta-cell preserving effect than the comparator. For the comparisons of PIR, however, when the calculated value was negative, the DPP-4 inhibitor was considered to have greater effect than the comparator.

Besides HOMA-β and PIR, mean values of patients' demographic data and clinical characteristics [including body mass index (BMI), glycosylated hemoglobin A1c (HbA1c) (%), and duration since diagnosis of type 2 DM] were curated for assessment of comparability of trials. Quality of study was assessed using the risk of bias tool provided by the Cochrane Handbook [27]. Two reviewers (MT and MS) independently evaluated all studies according to the following seven critical items: random sequence generation, allocation concealment, blinding of participants and personnel, blinding of outcome assessment, incomplete outcome data, selective reporting, and other bias (i.e., enough patients enrolled to reach the endpoint, adequate compliance). Publication bias was assessed visually using funnel plots and statistically using the method of Egger et al. [28]. Heterogeneity of curated data from individual studies was examined by Cochrane's Q test. When an $I^2$ value exceeded 50%, we considered that there was heterogeneity between the studies analyzed. We used fixed effects model when heterogeneity of curated data was rejected, and used a random effects model otherwise. For statistical analysis of meta-analysis, we calculated WMDs and their 95% CIs using Review Manager version 5.3 (The Nordic Cochrane Centre, the Cochrane Collaboration, Copenhagen, Denmark) based on an inverse variance method.

For sensitivity analyses, studies considered having high risk of bias in at least two out of seven domains (such as unblinded participants or personnel) were removed, and then analyses were repeated. In addition, we performed sensitivity tests when funnel plots of meta-analysis data suggested a possibility of publication bias. Specifically, we excluded suspected studies one by one from the corresponding meta-analysis and examined if there was a qualitative change in the result of statistical inference. When the statistical significance was lost by exclusion of a specific study, we considered that the result of the corresponding meta-analysis was conditional.

We presented the results of meta-analyses in a two-dimensional plot, in which we plotted the WMDs with 95% CIs obtained from comparisons of DPP-4 inhibitors versus other classes of hypoglycemia drugs (comparators: α-GIs, GLP-1 analogues, metformin, SGLT2 inhibitors, sulfonylureas, and thiazolidinediones) using HOMA-β (ordinate) and PIR (abscissa) as biomarkers. In addition, we depicted the approximate sample size of each comparison by the area of a circle in the plot.

## Results

### Characteristics of the included studies

In this meta-analysis, we retrieved 40 relevant studies from the three databases; MEDLINE, CENTRAL, and Ichushi-web, as shown in the flowchart (Fig 1). Among these studies, data for analysis were available for six of seven DPP-4 inhibitors: alogliptin (n = 2) [29, 30], anagliptin (n = 1) [31], linagliptin (n = 3) [32–34], saxagliptin (n = 4) [35–38], sitagliptin (n = 26) [39–

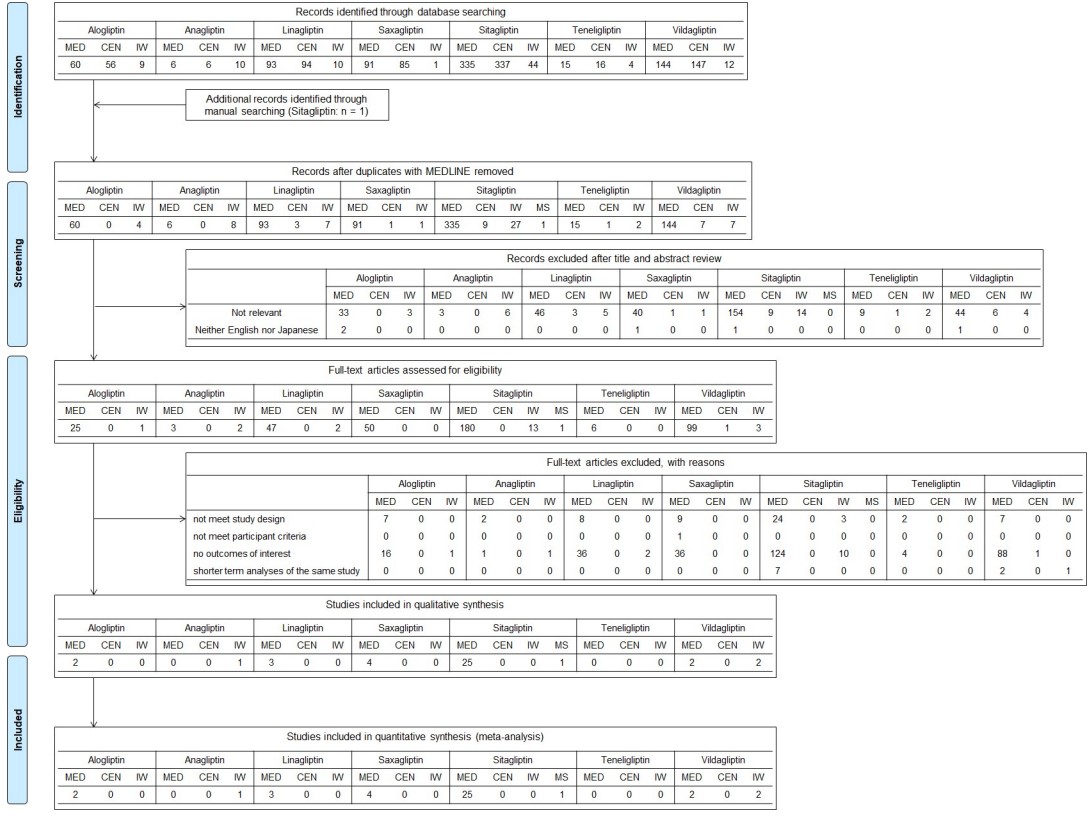

**Fig 1. Flowchart displaying the steps of selection of studies on seven dipeptidyl peptidase-4 inhibitors for meta-analysis.** Numbers of articles that met either the inclusion or exclusion criteria are shown in small tables. Abbreviations: CEN, CENTRAL; IW, Ichushi-web; MED, MEDLINE; MS, manual search.

64], and vildagliptin (n = 4) [65–68], but not for teneligliptin (n = 0). Comparators of DPP-4 inhibitors were α-GIs (n = 9) [30, 31, 34, 35, 41, 50, 63–65], GLP-1 analogues (n = 5) [47, 52, 58, 62, 68], metformin (n = 5) [29, 38–40, 56], SGLT2 inhibitors (n = 5) [36, 53, 54, 60, 61], sulfonylureas (n = 12) [32, 33, 37, 43–46, 48, 57, 59, 66, 67], and thiazolidinediones (n = 4) [42, 49, 51, 55].

A concise summary of the studies included in the meta-analysis was shown in Table 1. The overall mean HOMA-β at baseline was 55.9% (range: 25.5% to 150.6%), and mean PIR was 0.35 (range: 0.20 to 0.60). The mean follow-up period was 34 weeks (range: 8 to 104 weeks). As for the quality of studies, 11 studies [35, 38, 50, 52, 55, 57, 59, 61, 63, 66, 68] were considered at high risk of bias according to the tool of Cochrane Handbook. All 11 were open label studies (S3 Table). Particularly, two studies were considered to be at high risk of bias in three domains [50, 63]. Both studies compared the clinical outcomes between sitagliptin and α-GIs, and adherence was significantly lower in the α-GIs group than in the sitagliptin group.

## Comparison of DPP-4 inhibitors with other hypoglycemic drugs using HOMA-β as biomarker

Relevant data for comparing the effect on HOMA-β between DPP-4 inhibitors and other hypoglycemic drugs were available from eight studies of α-GIs [30, 31, 35, 41, 50, 63–65], five studies of GLP-1 analogues [47, 52, 58, 62, 68], five studies of metformin [29, 38, 39, 40, 56], five studies of SGLT2 inhibitors [36, 53, 54, 60, 61], ten studies of sulfonylureas [37, 43, 44–46,

**Table 1. Summary of studies included in the meta-analysis.**

| Study | Follow-up, weeks | Mean age, years | Mean BMI, kg/m² | Mean baseline HbA1c, % | Mean T2DM duration, years | Mean baseline HOMA-β % | Mean baseline PIR | Add-on therapy | DPP-4 inhibitor dose | No. of subjects in DPP-4 inhibitor group | Comparator drug dose | No. of subjects in comparator drug group | Outcome HOMA-β | Outcome PIR |
|---|---|---|---|---|---|---|---|---|---|---|---|---|---|---|
| Pratley 2014 [29] | 28 | 53.6 | 30.5 | NR | 3.7 | 71.3 | 0.30 | No | Alogliptin 25 mg qd | 112 | Metformin 500 mg bid | 114 | ✓ | ✓ |
| Seino 2011 [30] | 12 | 59.3 | 25.0 | 7.9 | 6.4 | NR | NR | No | Alogliptin 25 mg qd | 80 | Voglibose 0.2 mg tid | 83 | ✓ | |
| Kaku 2012 [31] | 12 | 59.0 | 24.6 | 7.7 | 8.0 | 30.6 | NR | No | Anagliptin 100 mg bid | 63 | Voglibose 0.2 mg tid | 65 | ✓ | ✓ |
| Forst 2010 [32] | 12 | 59.5 | 31.6 | 8.4 | 7.0 | NR | 0.20 | Metformin | Linagliptin 5 mg qd | 66 | Glimepiride 1–3 mg qd | 65 | | ✓ |
| Gallwitz 2012 [33] | 104 | 59.8 | 30.2 | 7.7 | > 5.0 (≥ 50% subjects) | NR | NR | Metformin | Linagliptin 5 mg qd | 776 | Glimepiride 1–4 mg qd | 775 | | ✓ |
| Kawamori 2012 [34] | 26 | 59.4 | 25.2 | 8.0 | > 5.0 (≥ 49% subjects) | NR | 0.27 | No | Linagliptin 5 mg qd | 159 | Voglibose 0.2 mg tid | 162 | | ✓ |
| Du 2017 [35] | 24 | 55.6 | 26.3 | 8.2 | 5.2 | 55.5 | NR | Metformin | Saxagliptin 5 mg qd | 238 | Acarbose 50–100 mg tid | 243 | ✓ | |
| Ekholm 2017 [36] | 24 | 54.5 | 31.5 | 8.9 | 7.7 | 40.3 | NR | Metformin | Saxagliptin 5 mg qd | 154 | Dapagliflozin 10 mg qd | 152 | ✓ | |
| Göke 2010 [37] | 52 | 57.6 | 31.4 | 7.7 | 5.4 | 67.5 | NR | Metformin | Saxagliptin 5 mg qd | 428 | Glipizide 5–20 mg qd | 430 | ✓ | |
| Tao 2018 [38] | 24 | 29.0 | 26.8 | 7.4 | 0 | 150.6 | NR | No | Saxagliptin 5 mg qd | 21 | Metformin 2000 mg qd | 21 | ✓ | ✓ |
| Aschner 2010 [39] | 24 | 56.0 | 30.8 | 7.2 | 2.4 | 85.4 | 0.31 | No | Sitagliptin 100 mg qd | 455 | Metformin 1000 mg bid | 439 | ✓ | ✓ |
| Berg 2011 [47] | 8 | 54.5 | 34.9 | 8.3 | 7.5 | 53.5 | NR | Metformin or thiazolidinediones | Sitagliptin 100 mg qd | 42 | Exenatide 5–10 μg bid | 41 | ✓ | |
| Derosa 2010 [40] | 52 | 57.5 | 27.8 | 8.5 | 5.5 | 53.3 | 0.38 | Pioglitazone | Sitagliptin 100 mg qd | 75 | Metformin 850 mg bid | 76 | ✓ | ✓ |
| Derosa 2013 [48] | 52 | NR | 27.5 | 7.2 | NR | 93.0 | 0.25 | Metformin + pioglitazone | Sitagliptin 100 mg qd | 228 | Glibenclamide 5 mg tid | 225 | ✓ | ✓ |
| Arjona 2013a [45] | 54 | 59.5 | 26.8 | 7.9 | 17.5 | NR | NR | No | Sitagliptin 25 mg qd | 64 | Glipizide 2.5 mg qd-10 mg bid | 65 | ✓ | ✓ |
| Arjona 2013b [46] | 54 | 64.5 | 26.8 | 7.8 | 10.4 | NR | NR | No | Sitagliptin 25–50 mg qd | 135 | Glipizide 2.5 mg qd-10 mg bid | 142 | ✓ | ✓ |
| Fukui 2015 [57] | 24 | 66.3 | 26.3 | 7.4 | 7.5 | 60.9 | 0.49 | Sulfonylurea | Sitagliptin 50 mg qd | 21 | Sulfonylurea dose not reported | 22 | ✓ | ✓ |
| Gadde 2017 [58] | 28 | 53.8 | 31.9 | 8.4 | 8.3 | 45.5 | NR | Metformin | Sitagliptin 100 mg qd | 122 | Exenatide 2.0 mg qw | 181 | ✓ | |
| Henry 2014 [49] | 54 | 51.8 | 31.3 | 8.8 | 4.1 | 68.7 | 0.37 | No | Sitagliptin 100 mg qd | 186 | Pioglitazone 45 mg qd | 188 | ✓ | ✓ |

*(Continued)*

**Table 1.** (Continued)

| Study | Follow-up, weeks | Mean age, years | Mean BMI, kg/m² | Mean baseline HbA1c, % | Mean T2DM duration, years | Mean baseline HOMA-β % | Mean baseline PIR | Add-on therapy | DPP-4 inhibitor dose | No. of subjects in DPP-4 inhibitor group | Comparator drug dose | No. of subjects in comparator drug group | Outcome HOMA-β | Outcome PIR |
|---|---|---|---|---|---|---|---|---|---|---|---|---|---|---|
| Iwamoto 2010 [41] | 12 | 60.7 | 24.7 | 7.8 | 5.2 | 35.5 | NR | No | Sitagliptin 50 mg qd | 163 | Voglibose 0.2 mg tid | 156 | ✓ | |
| Kobayashi 2014 [50] | 24 | 64.2 | 24.2 | 7.7 | 8.6 | 39.1 | NR | Glimepiride | Sitagliptin 50 mg qd | 59 | Voglibose 0.2 mg tid or miglitol 50 mg tid | 55 | ✓ | ✓ |
| Park 2017 [59] | 12 | 50.6 | 26.0 | 9.4 | 1.7 | 38.2 | NR | Metformin | Sitagliptin 100 mg qd | 21 | Glimepiride 2 mg qd | 21 | ✓ | |
| Pérez-Monteverde 2011 [51] | 12 | 51.1 | 29.8 | 9.1 | 3.2 | NR | NR | No | Sitagliptin 100 mg qd | 244 | Pioglitazone 15–30 mg qd | 248 | ✓ | ✓ |
| Pratley 2011 [52] | 52 | 55.5 | 32.6 | 8.5 | 6.2 | NR | NR | Metformin | Sitagliptin 100 mg qd | 219 | Liraglutide 1.2 mg qd | 225 | ✓ | ✓ |
| Pratley 2018 [60] | 26 | 55.1 | 31.6 | 8.6 | 6.8 | 50.4 | NR | Metformin | Sitagliptin 100 mg qd | 247 | Ertugliflozin 15 mg qd | 248 | ✓ | |
| Rosenstock 2012 [53] | 12 | 52.0 | 31.6 | 7.7 | 5.8 | 62.2 | NR | Metformin | Sitagliptin 100 mg qd | 65 | Canagliflozin 300 mg qd | 64 | ✓ | |
| Schernthaner 2013 [54] | 52 | 56.7 | 31.6 | 8.1 | 9.6 | 54.6 | NR | Metformin + sulfonylurea | Sitagliptin 100 mg qd | 378 | Canagliflozin 300 mg qd | 377 | ✓ | ✓ |
| Scott 2007 [43] | 12 | 54.9 | 30.5 | 7.9 | 4.5 | 51.9 | NR | OHA (except for thiazolidinediones) | Sitagliptin 50 mg bid | 124 | Glipizide 5–20 mg qd | 123 | ✓ | |
| Scott 2008 [42] | 18 | 55.0 | 30.3 | 7.8 | 4.8 | 62.7 | 0.34 | Metformin | Sitagliptin 100 mg qd | 94 | Rosiglitazone 8 mg qd | 87 | ✓ | ✓ |
| Seck 2010 [44] | 104 | 57.3 | 31.1 | 7.3 | 5.8 | 59.5 | 0.31 | Metformin | Sitagliptin 100 mg qd | 248 | Glipizide 5–20 mg qd | 256 | ✓ | ✓ |
| Shi 2019 [64] | 12 | 42.2 | 26.5 | 11.3 | 0 | NR | NR | Metformin + insulin | Sitagliptin 100 mg qd | 35 | Voglibose 0.2 mg tid | 35 | ✓ | |
| Takihata 2013 [55] | 24 | 60.5 | 25.2 | 7.4 | NR | 35.7 | NR | Metformin and/or sulfonylurea | Sitagliptin 50 mg qd | 58 | Pioglitazone 15 mg qd | 57 | ✓ | ✓ |
| Tsurutani 2018 [61] | 12 | 53.7 | 28.7 | 7.5 | 3 | 71.3 | NR | NR | Sitagliptin 50 mg qd | 59 | Ipragliflozin 50 mg qd | 60 | ✓ | |
| Weinstock 2015 [62] | 104 | 54.0 | 31.0 | 8.2 | 7.0 | NR | NR | Metformin and/or another OHA | Sitagliptin 100 mg qd | 315 | Duraglutide 0.75 mg qd | 302 | ✓ | ✓ |
| Williams-Herman 2010 [56] | 104 | 55.1 | 31.4 | 8.6 | 3.9 | 44.0 | 0.39 | No | Sitagliptin 100 mg qd | 52 | Metformin 500 mg bid | 65 | ✓ | ✓ |
| Yokoh 2015 [63] | 24 | 58.5 | 26.1 | 7.6 | 6.8 | 44.4 | NR | Metformin or pioglitazone | Sitagliptin 50 mg qd | 58 | Voglibose 0.2 mg tid or miglitol 50 mg tid | 58 | ✓ | ✓ |
| Iwamoto 2010 [65] | 12 | 59.1 | 24.9 | 7.6 | 5.4 | 31.7 | NR | No | Vildagliptin 50 mg bid | 188 | Voglibose 0.2 mg tid | 192 | ✓ | |

(Continued)

**Table 1.** (Continued)

| Study | Follow-up, weeks | Mean age, years | Mean BMI, kg/m² | Mean baseline HbA1c, % | Mean T2DM duration, years | Mean baseline HOMA-β % | Mean baseline PIR | Add-on therapy | DPP-4 inhibitor dose | No. of subjects in DPP-4 inhibitor group | Comparator drug dose | No. of subjects in comparator drug group | Outcome HOMA-β | PIR |
|---|---|---|---|---|---|---|---|---|---|---|---|---|---|---|
| Kim 2017 [66] | 12 | 56.0 | 25.9 | 7.6 | 6.1 | 55.5 | NR | Metformin | Vildagliptin 50 mg bid | 17 | Glimepiride 2 mg qd | 17 | ✓ | |
| Sawayama 2013 [67] | 26 | 68.4 | 26.2 | 7.4 | NR | 38.8 | NR | Glimepiride | Vildagliptin 50 mg qd | 11 | Glimepiride 1–2 mg qd | 12 | ✓ | |
| Takeshita 2015 [68] | 12 | 64.7 | 24.7 | 8.0 | NR | 25.5 | NR | Sitagliptin | Vildagliptin 50 mg bid | 58 | Liraglutide 0.3–0.9 mg qd | 54 | ✓ | |

Abbreviations: BMI, body mass index; DPP-4, dipeptidyl peptidase-4; HbA1c, glycosylated hemoglobin A1c; HOMA-β, homeostatic model of assessment for beta-cell function; NR, not reported; OHA, oral hypoglycemic agents; PIR, proinsulin-to-insulin ratio; T2DM, type 2 diabetes mellitus; qd, once daily; bid, twice daily; tid, three times daily; qw, once weekly.

48, 57, 59, 66, 67], and four studies of thiazolidinediones [42, 49, 51, 55] (Table 1). According to the results of Cochrane's Q test, the random effects model was employed for the meta-analysis of DPP-4 inhibitors versus α-GIs, SGLT2 inhibitors and sulfonylureas, but the fixed effects model was employed for DPP-4 inhibitors versus the remaining comparators. Analysis of all 37 studies showed that DPP-4 inhibitors increased HOMA-β to a significantly greater extent than α-GIs [WMD (95% CI): 7.54% (3.84, 11.24%)]. In contrast, DPP-4 inhibitors increased HOMA-β to a significantly less extent than GLP-1 analogues [-13.08% (-14.11, -12.05%)]. No significant differences in the change in HOMA-β were observed between DPP-4 inhibitors and metformin [-2.18% (-10.48, 6.12%)], between DPP-4 inhibitors and SGLT2 inhibitors [-4.08% (-11.65, 3.48%)], between DPP-4 inhibitors and sulfonylureas [-14.18% (-29.80, 1.45%)], and between DPP-4 inhibitors and thiazolidinediones [6.63% (-1.55, 14.81%)] (Fig 2).

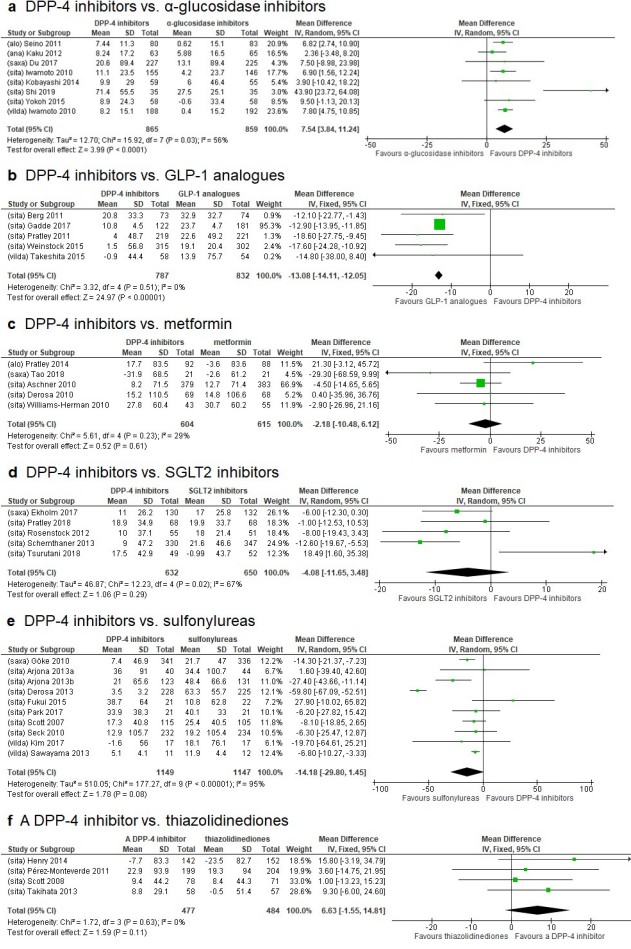

**Fig 2. Forest plots of meta-analyses for changes in homeostatic model of assessment for beta-cell function (HOMA-β).** Filled square represents weighted mean difference (WMD) of homeostatic model of assessment for beta-cell function (HOMA-β) for dipeptidyl peptidase-4 inhibitor versus other hypoglycemic agents. The two ends of the horizontal line passing through WMD represent the lower and upper limits of 95% confidence interval (CI). The size of each square corresponds to the weight assigned to the specific WMD data in the meta-analysis. Filled diamond at the bottom of each forest plot shows the combined WMD for individual comparison. Lateral tips of the diamond represent the lower and upper limits of 95% CI. A diamond located to the right of y-axis favors DPP-4 inhibitors over comparators. Abbreviations: CI, confidence interval; DPP-4, dipeptidyl peptidase-4; GLP-1, glucagon-like peptide-1; IV, inverse variance; SD, standard deviation; SGLT2, sodium-glucose cotransporter 2; alo, alogliptin; ana, anagliptin; saxa, saxagliptin; sita, sitagliptin; vilda, vildagliptin.

## Comparison of DPP-4 inhibitors with other hypoglycemic drugs using PIR as biomarker

Relevant data for comparing the effect on PIR between DPP-4 inhibitors and other hypoglycemic drugs were available from four studies of α-GIs [31, 34, 50, 63], one study of a GLP-1 analogue [52], four studies of metformin [29, 39, 40, 56], two studies of SGLT2 inhibitors [54, 61], seven studies of sulfonylureas [32, 33, 44–46, 48, 57], and three studies of thiazolidinediones [42, 49, 51] (Table 1). In accordance with the results of Cochrane Q test, the random effects model was employed for the comparison with sulfonylureas, and the fixed effects model for comparisons with all other drugs. Analysis of all 21 studies showed that DPP-4 inhibitors reduced PIR to a significantly greater extent than α-GIs [WMD (95%CI): -0.04 (-0.07, -0.02)] and SGLT2 inhibitors [-0.08 (-0.14, -0.03)]. In contrast, GLP-1 analogues and metformin reduced PIR to a significantly greater extent than DPP-4 inhibitors [0.06 (0.00, 0.12) and 0.04 (0.02, 0.06), respectively]. No statistically significant differences were detected for the comparisons of DPP-4 inhibitors with sulfonylureas [-0.02 (-0.05, 0.01)], and with thiazolidinediones [-0.01 (-0.05, 0.02)] (Fig 3).

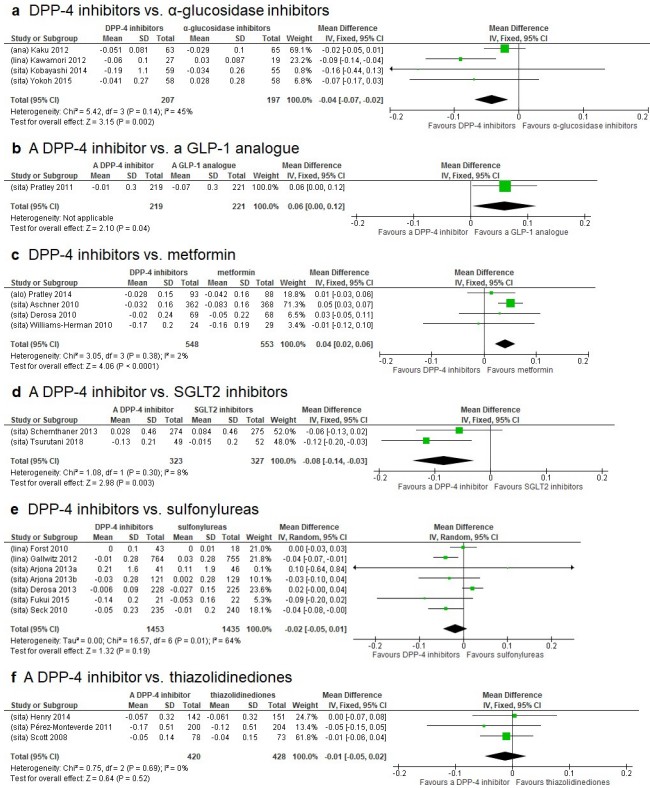

**Fig 3. Forest plots of meta-analyses for changes in proinsulin-to-insulin ratio (PIR).** Filled square represents weighted mean difference (WMD) of proinsulin-to-insulin ratio (PIR) for dipeptidyl peptidase-4 inhibitor versus other hypoglycemic agents. The two ends of the horizontal line passing through WMD represent the lower and upper limits of 95% confidence interval (CI). The size of each square corresponds to the weight assigned to the specific WMD data in the meta-analysis. Filled diamond at the bottom of each forest plot shows the combined WMD for individual comparison. Lateral tips of the diamond represent the lower and upper limits of 95% CI. A diamond located to the left of y-axis favors DPP-4 inhibitors over comparators. Abbreviations: CI, confidence interval; DPP-4, dipeptidyl peptidase-4; GLP-1, glucagon-like peptide-1; IV, inverse variance; SD, standard deviation; SGLT2, sodium-glucose cotransporter 2; alo, alogliptin; ana, anagliptin; lina, linagliptin; sita, sitagliptin.

## Two-dimensional display for the effects of DPP-4 inhibitors versus other hypoglycemic drugs using HOMA-β and PIR as biomarkers

Fig 4 showed a two-dimensional plot depicting WMDs with 95% CIs obtained from comparisons of DPP-4 inhibitors versus other classes of hypoglycemic drugs using HOMA-β and PIR as biomarkers. The data of each pair of comparison appeared as a cluster in one of the four quadrants. Numerical results of each comparison between DPP-4 inhibitors and other classes of hypoglycemia drugs already mentioned above. Fig 4 showed those results in an integrated manner to facilitate visual understanding. When both perpendicular and horizontal lines were present within the second quadrant, HOMA-β and PIR consistently showed superiority of DPP-4 inhibitors compared with other classes of hypoglycemic drugs. Whereas, when both perpendicular and horizontal lines were present within the fourth quadrant, HOMA-β and PIR consistently showed inferiority of DPP-4 inhibitors.

## Sensitivity analysis and evaluation of publication bias

We conducted a separate meta-analysis after excluding 11 studies [35, 38, 50, 52, 55, 57, 59, 61, 63, 66, 68], since they appeared to have high risk of bias according to Cochrane's bias tool (S3 Table). Excluding those studies changed the significance of the difference in HOMA-β for DPP-4 inhibitors versus SGLT2 inhibitors [-7.82% (-11.89, -3.75%)] and versus sulfonylureas [-18.70% (-36.86, -0.53%)].

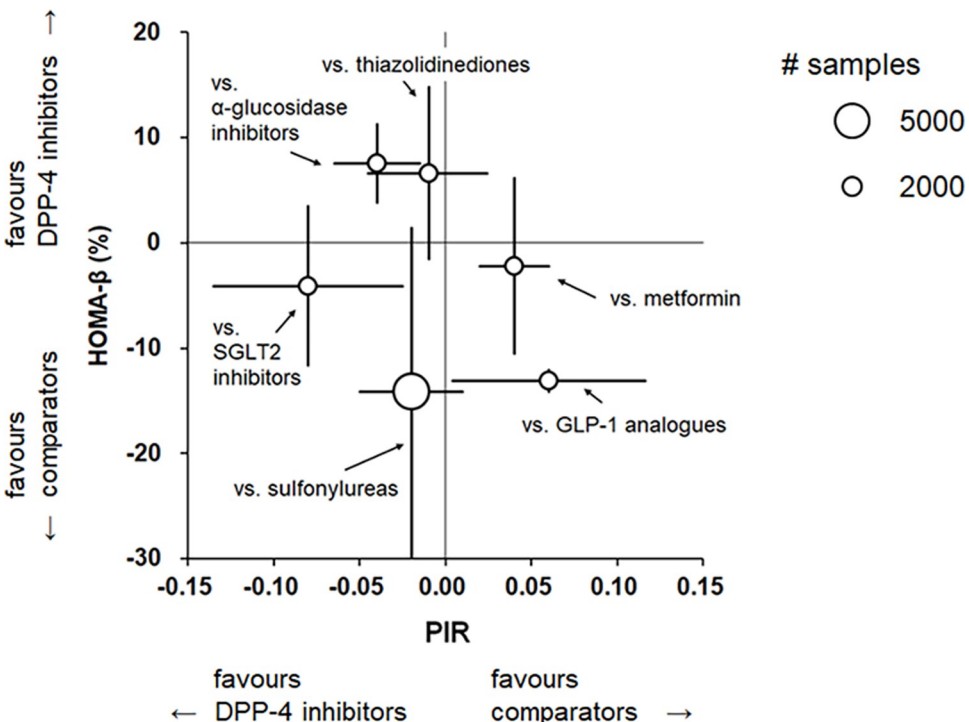

**Fig 4. Two-dimensional display of comparisons of DPP-4 inhibitors versus other hypoglycemic drugs using HOMA-β and PIR.** Combined weighted mean differences (WMDs) of homeostatic model of assessment for beta-cell function (HOMA-β) for various comparisons are plotted on the ordinate, and combined WMDs of proinsulin-to-insulin ratio (PIR) are plotted on the abscissa. The center of each circle represents the combined WMD of HOMA-β and PIR. Perpendicular and horizontal lines represent 95% confident intervals for HOMA-β and PIR, respectively. Approximate number of patients in the sample is shown by the area of the circle. Note that an increase in HOMA-β and a decrease in PIR are associated with amelioration of beta-cell function in patients with type 2 DM. Abbreviations: DPP-4, dipeptidyl peptidase-4; GLP-1, glucagon-like peptide-1; HOMA-β, homeostatic model of assessment for beta-cell function; PIR, proinsulin-to-insulin ratio; SGLT2, sodium-glucose cotransporter 2.

Funnel plots of the comparisons of DPP-4 inhibitors versus α-GIs, SGLT2 inhibitors, and sulfonylureas for HOMA-β (S1A, S1D and S1E Fig) as well as those of DPP-4 inhibitors versus α-GIs and sulfonylureas for PIR (S2A and S2D Fig) displayed apparently asymmetric distribution of data. However, the 95% CIs of the intercepts of Egger's regression lines [28] for HOMA-β between DPP-4 inhibitors and α-GIs [(-1.50, 3.47), p = 0.37], SGLT2 inhibitors [(-2.33, 9.37), p = 0.15] or sulfonylureas [(-5.71, 4.67), p = 0.82]; as well as for PIR between DPP-4 inhibitors and α-GIs [(-6.77, 3.72), p = 0.34] or sulfonylureas [(-4.27, 1.50), p = 0.27] included the origin of the coordinates. As a result, we concluded that sensitivity analysis of meta-analysis to evaluate the effect of publication bias may not be required for either HOMA-β or PIR.

## Discussion

To the best of our knowledge, the present study is the first to compare the pancreatic beta-cell preserving effect between DPP-4 inhibitors and other hypoglycemic drugs using two distinct biomarkers, HOMA-β and PIR. In this study, the results of HOMA-β and PIR consistently showed superiority of DPP-4 inhibitors over α-GIs and inferiority of DPP-4 inhibitors to GLP-1 analogues. However, PIR showed inferiority of DPP-4 inhibitors to metformin and superiority over SGLT2 inhibitors, while HOMA-β showed no significant differences in these comparisons. On the other hand, no significant differences in the change in both HOMA-β and PIR were observed between DPP-4 inhibitors and sulfonylureas as well as between DPP-4 inhibitors and thiazolidinediones. Recently, Wu et al. [69] reported a network meta-analysis on the comparisons of incretin-based therapies (i.e. DPP-4 inhibitors and GLP-1 analogues) with α-GIs, metformin, sulfonylureas, thiazolidinediones and placebo using HOMA-β as the sole biomarker for beta-cell function in patients with type 2 DM. Their results agree well with our present meta-analysis. The implication of our study is that results of comparison of beta-cell preserving effect between DPP-4 inhibitors and other hypoglycemic drugs using HOMA-β and PIR as biomarker should be interpreted with caution, because the two biomarkers may indicate different functions of beta-cell.

The changes observed in WMDs for HOMA-β and PIR yielded comparable results in the comparisons between DPP-4 inhibitors and two hypoglycemic drugs; α-GIs and GLP-1 analogues, in patients with type 2 DM. These results suggest that DPP-4 inhibitors are superior to α-GIs but inferior to GLP-1 analogues in terms of beta-cell preserving effect. In our meta-analysis between DPP-4 inhibitors and α-GIs using all the retrieved articles, we included a study in which all participants were given insulin [64]. Nevertheless, removal of that study [64] from meta-analysis did not change the statistical results.

On the other hand, the changes observed in WMDs for HOMA-β and PIR when comparing DPP-4 inhibitors and sulfonylureas showed opposite trends: the changes in HOMA-β appeared to favor sulfonylureas over DPP-4 inhibitors, whereas those in PIR appeared opposite. Hyperglycemia in patients with type 2 DM may be attributed to reduced insulin secretion from the pancreas and/or to peripheral insulin resistance. Attenuated basal secretion of insulin which is stimulated mainly by fasting serum glucose levels can be detected by HOMA-β [70]. However, insulin secretion in patients receiving sulfonylureas may be augmented even during euglycemia by the pharmacological effects of the drugs [71]. As a result, increases of HOMA-β observed after the initiation of sulfonylurea may not be attributed simply to the beta-cell preserving effect of the drug [72]. In our study, the comparison between DPP-4 inhibitors and sulfonylureas using HOMA-β as biomarker did not reach a significant level. One of the reasons for this result may be the selection of statistical model. We employed the random effect model, since the data reported by Derosa et al. [48] showed a heterogenously large effect on HOMA-

β. Indeed, when their data were removed from the comparative integration analysis, heterogeneity disappeared and the overall comparison was statistically significant. The heterogenous data of Derosa et al. were probably due to markedly higher baseline HOMA-β in their patients (93.0%) compared to other studies used in comparison (range: 38.2% to 67.5%). In contrast, there was no significant difference in baseline PIR between the study of Derosa et al (0.25) and the other studies (range: 0.20 to 0.49). PIR is associated with the efficiency of proinsulin processing to insulin within beta-cells rather than the beta-cell mass [12]. In this respect, PIR may be a more suitable biomarker than HOMA-β for assessing the beta-cell function in patients with type 2 DM receiving sulfonylureas. In the present study, no significant differences were observed between DPP-4 inhibitors and sulfonylureas when using HOMA-β or PIR, although there was a trend of superiority for DPP-4 inhibitors over sulfonylureas when using PIR.

HOMA-β and PIR showed discrepant results in the assessment of beta-cell preserving effects of SGLT2 inhibitors. HOMA-β was proposed as a marker of beta-cell function on the assumption that blood glucose level depends only the rate of insulin secretion rate [22]. In patients receiving SGLT2 inhibitors, however, blood glucose levels are controlled not only by the insulin-dependent intracellular transport but also by the SGLT2 inhibitor-induced augmentation of urinary loss of glucose [73]. Consequently, the effect of SGLT2 inhibitors on beta-cell preservation may be overestimated by HOMA-β. The other biomarker examined in this study, PIR, was employed in a recent large cohort study searching for clinical factors that influence the progression of pancreatic beta-cell dysfunction (BetaDecline study) [74]. Taken together, PIR may be a useful biomarker of pancreatic beta-cell function for comparing DPP-4 inhibitors with SGLT2 inhibitors.

There are few studies on the effects of thiazolidinediones and metformin on beta-cell function in humans. Ishida et al. [75] reported that pioglitazone enhanced beta-cell function in diabetic mice through reduction of oxidative stress in pancreatic beta-cells. It has been reported that the reduction of oxidative stress could be due to increase the intrinsic activity of the glucose transporters [76] or augmented beta-cell pancreatic duodenal homeobox-1 expression [77]. Meanwhile, the favorable effects of metformin for beta-cell function are considered to be partially based on improving insulin processing insufficiency. Nagi et al. [78] reported that metformin decreased plasma proinsulin concentrations in subjects with type 2 diabetes mellitus. Their findings support the results of our meta-analysis that metformin reduced PIR to a significantly greater extent than DPP-4 inhibitors. However, precise mechanisms by which thiazolidinediones and metformin improve beta-cell function in clinical settings remain to be clarified.

In the present study, we made a new comprehensive attempt to compare the beta-cell preserving effects of hypoglycemic drugs using HOMA-β and PIR as biomarkers. We present a two-dimensional display of the beta-cell preserving effect of DPP-4 inhibitors versus other hypoglycemic drugs when assessed by HOMA-β and PIR (Fig 4). While an increase in HOMA-β may be associated with amelioration of the beta-cell function of insulin secretion [11], a reduction in PIR may be related to the improvement of proinsulin processing to insulin [79]. Therefore, data plotted in either the second or fourth quadrant imply qualitative similarity of HOMA-β and PIR in assessing the beta-cell function preserving effect of hypoglycemic drugs. We revealed that there is an agreement between HOMA-β and PIR regarding the pancreatic beta-cell preserving effects of hypoglycemic drugs compared with DPP-4 inhibitors, except for sulfonylureas and SGLT2 inhibitors for which the assumption of HOMA-β may not hold. In the plot of HOMA-β versus PIR in Fig 4, comparisons between DPP-4 inhibitors and sulfonylureas or SGLT2 inhibitors are plotted in the third quadrant. Therefore, evaluation using HOMA-β alone might not be suitable. Data presentation as Fig 4 may be useful for comprehensive interpretation of the pancreatic beta cell function preserving effects of

hypoglycemic drugs with different mechanisms of action. Further studies using this approach to compare the performance of different biomarkers of pancreatic beta-cells are anticipated.

HOMA-β is most frequently used as a clinical and epidemiological biomarker for assessing pancreatic beta-cell function. However, our results of comparison between DPP-4 inhibitors and sulfonylureas or SGLT2 inhibitors illustrate that HOMA-β may not reflect beta-cell function accurately for drugs with hypoglycemic actions via direct insulin secretion or independent of insulin secretion. In addition, although HOMA-β assumes that all glucose-lowering action is provided by correctly processed insulin derived from beta-cells [80], impaired processing of proinsulin to insulin is observed in patients with type 2 DM [81].

One of the limitations of the present study is inherent to the methodology of meta-analysis. The biomarkers employed for evaluating the functional beta-cell reserve; HOMA-β and PIR, were secondary endpoints of the clinical trials included in the present meta-analysis. While the Cochrane risk of bias tool may be useful for assessing the quality of studies and the primary outcome, whether it is also valid for assessing the quality of the secondary outcome remains unclear. In particular, when the data of secondary outcomes are derived from a subgroup of patients, random allocation of patients may not be guaranteed. As a result, we cannot categorically eliminate selection bias or attrition bias for the assessment of the secondary outcomes analyzed in the present study. In addition, we cannot eliminate publication bias legitimately from meta-analysis despite the performance of funnel plot analyses. Furthermore, Egger's regression analysis used for assessing bias and homogeneity of studies may be useful if more than ten publications are available for each integration analysis [82].

## Conclusions

We performed comprehensive meta-analyses of the effects of DPP-4 inhibitors on pancreatic beta-cell function compared with other classes of hypoglycemic drugs using HOMA-β and PIR as biomarkers. DPP-4 inhibitors appear to be superior to α-GIs but inferior to GLP-1 analogues in terms of preservation of beta-cell function assessed by either HOMA-β or PIR. DPP-4 inhibitors seem to be superior to SGLT2 inhibitors but inferior to metformin in terms of islet function assessed only by PIR. Since HOMA-β and PIR may represent different aspects of beta-cell function, results of comparison of beta-cell function preserving effects among hypoglycemic agents should be interpreted with caution.

## Supporting information

**S1 Fig. Funnel plots of the data employed in the meta-analysis of homeostatic model of assessment for beta-cell function (HOMA-β).** This plot is effect size (weighted mean difference of homeostatic model of assessment for beta-cell function [HOMA-β] from individual articles) (X axis) versus standard error of effect size (Y axis). Dotted vertical line represents pooled estimate of effects. Abbreviations: DPP-4, dipeptidyl peptidase-4; GLP-1, glucagon-like peptide-1; MD, mean difference; SE, standard error; SGLT2, sodium-glucose cotransporter 2. (TIF)

**S2 Fig. Funnel plots of the data employed for the meta-analysis of proinsulin-to-insulin ratio (PIR).** This plot is effect size (weighted mean difference of proinsulin-to-insulin ratio [PIR] from individual articles) (X axis) versus standard error of effect size (Y axis). Dotted vertical line represents pooled estimate of effects. Abbreviations: DPP-4, dipeptidyl peptidase-4; MD, mean difference; SE, standard error; SGLT2, sodium-glucose cotransporter 2. (TIF)

**S1 Table. PRISMA 2009 checklist.**
(DOC)

**S2 Table. Detailed search strategies for the MEDLINE, CENTRAL and Ichushi-web databases.**
(DOCX)

**S3 Table. Summary of the authors' judgement on the risk of bias of the publications included in the present study.** Symbols: +, low risk of bias; -, high risk of bias; ?, unclear risk of bias.
(DOCX)

## Author Contributions

**Conceptualization:** Masahiro Takahashi, Hirotoshi Echizen.

**Data curation:** Masahiro Takahashi, Misa Shibasaki.

**Formal analysis:** Masahiro Takahashi, Misa Shibasaki.

**Investigation:** Masahiro Takahashi, Misa Shibasaki.

**Methodology:** Masahiro Takahashi.

**Project administration:** Masahiro Takahashi.

**Supervision:** Masahiro Takahashi.

**Validation:** Masahiro Takahashi, Misa Shibasaki.

**Visualization:** Masahiro Takahashi, Misa Shibasaki.

**Writing – original draft:** Masahiro Takahashi.

**Writing – review & editing:** Hirotoshi Echizen, Akifumi Kushiyama.

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
