## [Decision Letter · Decision Letter 0]

29 May 2020

PONE-D-20-13427

Comparisons between dipeptidyl peptidase-4 inhibitors and other classes of hypoglycemic drugs using two distinct biomarkers of pancreatic beta-cell function: a meta-analysis

PLOS ONE

Dear Dr. Takahashi,

Thank you for submitting your manuscript to PLOS ONE. After careful consideration, we feel that it has merit but does not fully meet PLOS ONE’s publication criteria as it currently stands. Therefore, we invite you to submit a revised version of the manuscript that addresses the points raised during the review process.

We look forward to receiving your revised manuscript.

Kind regards,

Tatsuo Shimosawa, M.D., Ph.D.

Academic Editor

PLOS ONE

2. Please ensure you have included the full electronic search strategy for at least one database and uploaded it as an additional file.

Reviewers' comments:

Reviewer's Responses to Questions

**Comments to the Author**

1. Is the manuscript technically sound, and do the data support the conclusions?

Reviewer #1: No

Reviewer #2: Partly

2. Has the statistical analysis been performed appropriately and rigorously? 

Reviewer #1: I Don't Know

Reviewer #2: N/A

3. Have the authors made all data underlying the findings in their manuscript fully available?

Reviewer #1: Yes

Reviewer #2: Yes

4. Is the manuscript presented in an intelligible fashion and written in standard English?

Reviewer #1: Yes

Reviewer #2: No

5. Review Comments to the Author

Reviewer #1: In the manuscript entitled "Comparisons between dipeptidyl peptidase-4 inhibitors and other classes of hypoglycemic drugs using two distinct biomarkers of pancreatic beta-cell function: a

meta-analysis", authors compared, by meta-analysis of published literatures, effects of DPP-4 inhibitors and other classes of hypoglycemic drugs on HOMA-beta and proinsulin-to-insulin ratio (PIR). They found that DPP-4i is superior to aGI but inferior to GLP-1RA regarding both HOMA and PIR. In addition, PIR is found to show inferiority of DPP-4 inhibitors compared with metformin, and superiority compared with SGLT2 inhibitors, while there are no differences between DPP4i and met or SGLT2i.

Analysis has been well conducted and the reviewer agrees with their main conclusion. My concern is about use of HOMA parameters in general. As discussed in ref. 8, HOMA-beta shows activity of beta-cells but never helth status of beta-cell. An increased HOMA-beta in SU users is sometimes mistakenly interpreted. The data in this MS clearly showed this fact. Authors should discuss the abuse of HOMA modeling in DISCUSSION.

Reviewer #2: This study conducted a meta-analysis to compare the effects of DPP-4 inhibitors and other classes of hypoglycemic drugs on HOMA-β and proinsulin-to-insulin ratio (PIR). The main finding included that HOMA-β and PIR consistently showed superiority of DPP-4 inhibitors compared with α-GI; both biomarkers consistently supported inferiority of DPP-4 inhibitors compared with GLP-1 analogues; PIR showed inferiority of DPP-4 inhibitors compared with metformin, and superiority compared with SGLT2 inhibitors, whereas HOMA-β showed no significant differences between DPP-4 inhibitors and the two others. I have several suggestions for the authors to consider:

1. As described in the abstract, the aim of this study was to compare the effects of DPP-4 inhibitors and other classes of hypoglycemic drugs on HOMA-β and proinsulin-to-insulin ratio (PIR). However, the conclusion highlighted the application of HOMA-β and PIR rather than the special effects of DPP-4 on HOMA-β and PIR relative to other drugs. The conclusion seems to be beyond the major purpose of the study.

2. Why didn’t authors include the Embase in the literature search?

3. Please show the detailed search strategies.

4. Why didn’t authors exclude the study in which background medications include insulin? For example, Shi C 2019 (Reference 64). The use of exogenous insulin may lead to incorrect values of HOMA-β and PIR.

5. In the introduction, authors posed two questions: whether these incretin-related drugs have advantage over other hypoglycemic drugs by exhibiting pancreatic beta-cell function preserving effect; whether there are different effects of beta-cell function preservation between DPP-4 inhibitors and other classes of hypoglycemic drugs using both HOMA-β and PIR as biomarkers. In the discussion, however, authors mainly discussed the advantage and/or disadvantage of HOMA-β and PIR as the biomarkers to assess the effect of various hypoglycemic agents on the beta-cell function in the discussion. Therefore, has the study already answered the questions posed in the introduction?

6. What is the criterion to determine whether or not HOMA-β or PIR is a suitable biomarker for comparisons of beta-cell function preserving effect between hypoglycemic drugs with different mechanisms of action?

7. According to the meta-analysis, if HOMA-β and PIR are used together to assess the effect of various hypoglycemic drugs on beta-cell function preservation, how would the results be explained? Namely, should be the results of HOMA-β and PIR consistent or not? If consistent, what does it suggest? If inconsistent, what does it suggest?

For different hypoglycemic drugs, how to appropriately use HOMA-β and PIR?

8. English needs revision. Some linguistic errors have been made. A final check of the language by a native English speaking person may lead to the necessary improvements.

6. PLOS authors have the option to publish the peer review history of their article (what does this mean?). If published, this will include your full peer review and any attached files.

Reviewer #1: No

Reviewer #2: No

---

## [Author Response · Author response to Decision Letter 0]

30 Jun 2020

Authors’ responses to Reviewer #1 and Reviewer #2

1. We have made our responses to reviewers’ comments and indicated their corresponding revisions in text with page and line numbers in “Revised Article with Changes Highlighted” file.

2. For responding to the comment 3 from Reviewer #2, we have added the data as a S2 Table. As a result, we have renumbered supplemental Tables.

3. According to the revisions made in the original manuscript, the reference 80 and subsequent ones were renumbered.

4. Pages in S1 Table were renumbered, according to the revisions made in the manuscript.

5. Except for page and line numbers shown below, we have revised the manuscript according to proofreading.

6. We have used the following abbreviations in this text: DPP-4, dipeptidyl peptidase-4; HOMA-β, homeostatic model of assessment for beta-cell function; PIR, proinsulin-to-insulin ratio; SGLT2, sodium-glucose cotransporter 2

Authors’ response to Reviewer #1:

Comment: My concern is about use of HOMA parameters in general. As discussed in ref. 8, HOMA-beta shows activity of beta-cells but never health status of beta-cell. An increased HOMA-beta in SU users is sometimes mistakenly interpreted. The data in this MS clearly showed this fact. Authors should discuss the abuse of HOMA modeling in DISCUSSION.

Response: We thank Reviewer #1’s important comment. According to his/her comment, we have added the explanations about the abuse of HOMA-β in general citing comparative results between DPP-4 inhibitors and sulfonylureas or SGLT2 inhibitors as examples (pp. 28-29, lines 423-430).

Authors’ responses to Reviewer #2:

We deeply appreciate Reviwer #2’s comments. We would like to answer to his/her comments as follows.

Comment 1: As described in the abstract, the aim of this study was to compare the effects of DPP-4 inhibitors and other classes of hypoglycemic drugs on HOMA-β and PIR. However, the conclusion highlighted the application of HOMA-β and PIR rather than the special effects of DPP-4 on HOMA-β and PIR relative to other drugs. The conclusion seems to be beyond the major purpose of the study.

Response: We agree with Reviewer #2’s comment. We have revised the conclusion of abstract so that we have described the answer corresponding to the aim of the study (p. 3, lines 41-46). In addition, we have revised the discussion and conclusion of the main text to clarify the point of our study (pp. 23-24, lines 333-336; p. 24, lines 343-345; p. 26, lines 377-378; pp. 26-27, lines 387-388; p. 30, lines 448-454).

Comment 2: Why didn’t authors include the Embase in the literature search?

Response: For reducing potential biases (mainly selection bias), we performed literature search of meta-analysis using multiple databases. MEDLINE and EMBASE are two major databases for systematic literature search. Halladay CW (J Clin Epidemiol. 2015;68(9):1076) and Slobogean GP (J Clin Epidemiol. 2009;62(12):1261) reported that EMBASE may give additional information to those of MEDLINE. Slobogean also reported that the CENTRAL database may be useful for increasing relevant literature in addition to MEDLINE and EMBASE. Therefore, we have recognized that we could perform a systematic literature search through using MEDLINE, CENTRAL and Ichushi-web. 

Comment 3: Please show the detailed search strategies.

Response: We have added the detailed search strategies for the MEDLINE, CENTRAL and Ichushi-web databases in S2 Table. In addition, we have described these revisions in text (p. 6, lines 90-91).

Comment 4: Why didn’t authors exclude the study in which background medications include insulin? For example, Shi C 2019 (Reference 64). The use of exogenous insulin may lead to incorrect values of HOMA-β and PIR.

Response: As Reviewer #2 pointed out, the administration of exogenous insulin (and other medications) may influence on HOMA-β and PIR values. However, we performed our meta-analysis with studies performed with the randomized design to even out the impact of the bias deriving from various background medications. It was difficult to eliminate or evaluate completely the effects of exogenous insulin or other medications, because the curated studies did not report all the medications including insulin for rescue use. Furthermore, we found little impact on the results of our study by removing the Reference 64: WMD of HOMA-β [95% CI] for the original data vs. that without the Reference 64 were 7.54% [3.84, 11.24%] and 6.78% [4.78, 8.78%], respectively. In this context, we have added a comment about influence of exogenous insulin to our results (p. 24, lines 345-348).

Comment 5: In the introduction, authors posed two questions: whether these incretin-related drugs have advantage over other hypoglycemic drugs by exhibiting pancreatic beta-cell function preserving effect; whether there are different effects of beta-cell function preservation between DPP-4 inhibitors and other classes of hypoglycemic drugs using both HOMA-β and PIR as biomarkers. In the discussion, however, authors mainly discussed the advantage and/or disadvantage of HOMA-β and PIR as the biomarkers to assess the effect of various hypoglycemic agents on the beta-cell function in the discussion. Therefore, has the study already answered the questions posed in the introduction?

Response: We agree with Reviewer #2 that we did not answered directly the questions we posed in the introduction. According to the comment, we have revised the first paragraph of introduction section so that we clearly described that the aim of the present study was to compare the effects of DPP-4 inhibitors with other hypoglycemic drugs (p. 4, lines 55-61, including revision of Reference 5-7). We also agree with the other comment that the discussion about advantage and disadvantage of HOMA-β and PIR as biomarkers of the pancreatic beta-cell function may be beyond the range of the present study. We might have discussed too much about the physiological aspects of the two biomarkers. However, we consider that our descriptions would have been needed for facilitating understanding our data to general readers who do not have deep knowledge about the two biomarkers.

Comment 6: What is the criterion to determine whether or not HOMA-β or PIR is a suitable biomarker for comparisons of beta-cell function preserving effect between hypoglycemic drugs with different mechanisms of action?

Response: We agree with Reviewer #2 that the criterion might be useful to determine either HOMA-β or PIR as a better biomarker for assessing the pancreatic beta-cell preserving effects of different hypoglycemic drugs. We do not have a comprehensive and absolute criterion about the choice of biomarkers, since we performed only pair-wise comparison in the present study. Instead, we made Fig 4 for helping the understanding of our data. Inconsistent results between HOMA-β and PIR may be useful information about the two biomarkers. For example, we may say that HOMA-β overestimates the effects of sulfonylureas (p. 25, lines 354-359) and SGLT2 inhibitors (p. 26, lines 379-384) on beta-cell function, and PIR would be a useful biomarker for these drugs.

Comment 7: According to the meta-analysis, if HOMA-β and PIR are used together to assess the effect of various hypoglycemic drugs on beta-cell function preservation, how would the results be explained? Namely, should be the results of HOMA-β and PIR consistent or not? If consistent, what does it suggest? If inconsistent, what does it suggest? For different hypoglycemic drugs, how to appropriately use HOMA-β and PIR?

Response: We agree with Reviewer #2’s thoughtful comment. We consider that secretion of insulin assessed by HOMA-β and processing of proinsulin assessed by PIR may represent different aspects of beta-cell functions (pp. 4-5, lines 66-68). The selection of one biomarker over the other depends on the mechanism of hypoglycemic action of drugs to be compared. If the results of HOMA-β and PIR are consistent, we may say the advantage of one drug over the other comprehensively (p. 24, lines 343-345). However, if the results are inconsistent, we may look for the reasons according to the pharmacological mechanisms of the respective drugs (pp. 26-27, lines 376-388; p. 27, lines 396-403).

Comment 8: English needs revision. Some linguistic errors have been made. A final check of the language by a native English speaking person may lead to the necessary improvements.

Response: We have asked the proofreading of the revised manuscript to another native English speaker.

---

## [Decision Letter · Decision Letter 1]

10 Jul 2020

Comparisons between dipeptidyl peptidase-4 inhibitors and other classes of hypoglycemic drugs using two distinct biomarkers of pancreatic beta-cell function: a meta-analysis

PONE-D-20-13427R1

Dear Dr. Takahashi,

We’re pleased to inform you that your manuscript has been judged scientifically suitable for publication and will be formally accepted for publication once you change Line 344:  Replacing 'pancreatic preserving effect' with 'beta-cell preserving effect’ and it meets all outstanding technical requirements.

Kind regards,

Tatsuo Shimosawa, M.D., Ph.D.

Academic Editor

PLOS ONE

Additional Editor Comments (optional):

Reviewers' comments:

Reviewer's Responses to Questions

**Comments to the Author**

1. If the authors have adequately addressed your comments raised in a previous round of review and you feel that this manuscript is now acceptable for publication, you may indicate that here to bypass the “Comments to the Author” section, enter your conflict of interest statement in the “Confidential to Editor” section, and submit your "Accept" recommendation.

Reviewer #1: (No Response)

Reviewer #2: All comments have been addressed

2. Is the manuscript technically sound, and do the data support the conclusions?

Reviewer #1: (No Response)

Reviewer #2: Yes

3. Has the statistical analysis been performed appropriately and rigorously? 

Reviewer #1: (No Response)

Reviewer #2: Yes

4. Have the authors made all data underlying the findings in their manuscript fully available?

Reviewer #1: (No Response)

Reviewer #2: Yes

5. Is the manuscript presented in an intelligible fashion and written in standard English?

Reviewer #1: (No Response)

Reviewer #2: Yes

6. Review Comments to the Author

Reviewer #1: (No Response)

Reviewer #2: The authors did a good job and further improved the manuscript in which the reviewer’s comments were adequately answered. A minor revision was suggested as below:

Line 344: Suggest replacing 'pancreatic preserving effect' with 'beta-cell preserving effect’.

7. PLOS authors have the option to publish the peer review history of their article (what does this mean?). If published, this will include your full peer review and any attached files.

Reviewer #1: No

Reviewer #2: No

---

## [Editor Report · Acceptance letter]

14 Jul 2020

PONE-D-20-13427R1 

Comparisons between dipeptidyl peptidase-4 inhibitors and other classes of hypoglycemic drugs using two distinct biomarkers of pancreatic beta-cell function: a meta-analysis 

Dear Dr. Takahashi:

I'm pleased to inform you that your manuscript has been deemed suitable for publication in PLOS ONE. Congratulations! Your manuscript is now with our production department. 

Kind regards, 

on behalf of

Prof. Tatsuo Shimosawa 

Academic Editor

PLOS ONE